# Multifunctional Oxidized Succinoglycan/Poly(N-isopropylacrylamide-co-acrylamide) Hydrogels for Drug Delivery

**DOI:** 10.3390/polym15010122

**Published:** 2022-12-28

**Authors:** Yiluo Hu, Younghyun Shin, Sohyun Park, Jae-pil Jeong, Yohan Kim, Seunho Jung

**Affiliations:** 1Department of Bioscience and Biotechnology, Microbial Carbohydrate Resource Bank (MCRB), Konkuk University, Seoul 05029, Republic of Korea; 2Department of Systems Biotechnology, Microbial Carbohydrate Resource Bank (MCRB), Konkuk Univesity, Seoul 05029, Republic of Korea

**Keywords:** succinoglycan, poly(N-isopropylacrylamide-co-acrylamide), temperature/pH-responsive hydrogel, stimuli-responsive drug delivery

## Abstract

We prepared the self-healing and temperature/pH-responsive hydrogels using oxidized succinoglycan (OSG) and a poly (N-isopropyl acrylamide-co-acrylamide) [P(NIPAM-AM)] copolymer. OSG was synthesized by periodate oxidation of succinoglycan (SG) isolated directly from soil microorganisms, *Sinorhizobium meliloti* Rm1021. The OSG/P(NIPAM-AM) hydrogels were obtained by introducing OSG into P(NIPAM-AM) networks. The chemical structure and physical properties of these hydrogels were characterized by ATR-FTIR, XRD, TGA, and FE-SEM. The OSG/P(NIPAM-AM) hydrogels showed improved elasticity, increased thermal stability, new self-healing ability, and 4-fold enhanced tensile strength compared with the P(NIPAM-AM) hydrogels. Furthermore, the 5-FU-loaded OSG/P(NIPAM-AM) hydrogels exhibited effective temperature/pH-responsive drug release. Cytotoxicity experiments showed that the OSG/P(NIPAM-AM) hydrogels were non-toxic, suggesting that OSG/P(NIPAM-AM) hydrogels could have the potential for biomedical applications, such as stimuli-responsive drug delivery systems, wound healing, smart scaffolds, and tissue engineering.

## 1. Introduction

Hydrogel, a three-dimensional network structure cross-linked with polymer chains, has a porous form and super absorbency, depending on the polymer properties [1]. Stimulus-responsive hydrogels that can change their volume, shape, and swelling behavior in response to external physical or chemical stimuli are advanced biotechnology materials that can be used in various applications, including tissue engineering and biomedical and drug delivery systems [2,3]. Each of the environmental stimulus responsivity can be classified into pH, temperature, light, redox reaction, chemicals, electric/magnetic fields, etc. [4,5,6]. Among them, pH- or temperature-sensitive hydrogels are the most extensively studied polymer systems [7]. Poly(N-Isopropylacrylamide) (PNIPAM), one of the thermally reactive polymers, has been the most studied because it exhibits a unique volume phase transition from the hydrolyzed expansion state to the collapsed state at a low critical solution temperature (LCST) of about 32 °C [6,8]. In contrast, acrylamide and acrylic acid are typical examples of pH-responsive synthetic polymers [9,10]. Many studies have been conducted on temperature/pH dual-sensitive hydrogels by combining the thermo-sensitivity of PNIPAM with the pH-sensitivity of poly(acrylamide) (PAM) [11,12]. PNIPAM is a synthetic polymer composed of an aqueous amide (-CONH-) and a hydrophobic isopropyl (-CH(CH_3_)_2_) side chain group. PNIPAM-based hydrogels are inherently temperature-responsive and exhibit rapid swelling/de-swelling near a specific temperature called the volume phase transition temperature [13]. This reversible change process is because PNIPAM exhibits hydrophobicity when the temperature of the external environment is higher than LCST and hydrophilicity when the temperature is lower than LCST. PAM-based hydrogels have a high water content, can keep the wound moist, and promote the maintenance of a normal physiological environment, which helps in tissue wound healing [14]. However, those synthetic polymer hydrogels have limited biocompatibility and mechanical properties.

Succinoglycan (SG) is a rhizobial exopolysaccharide produced by some soil bacteria, such as *Agrobacterium*, *Rhizobium*, and *Sinorhizobium* species [15,16]. SG is an anionic polysaccharide with eight sugar repeating units, seven glucose, and one galactose residues linked to each other to form a succinyl group, a pyruvyl group, and an acetyl group [17]. SG has been commonly applied in food and cosmetic industries because of its high viscosity, thermal stability, emulsifying activity, and plasticizing-like properties [18,19]. The iodide oxidation of SG served as a novel support to form Schiff bases with the amine groups of cross-linked polymers by introducing aldehyde groups into the crosslinking networks. Recent reports have shown that OSG-based hydrogels have improved physical and mechanical properties [20,21].

This study introduced OSG into a P(NIPAM-AM) copolymer network to fabricate a temperature/pH dual-sensitive hydrogel with high biocompatibility and excellent physical and mechanical properties. The manufacturing process is specified in Figure 1, and the research hypothesis for this hydrogel was established as follows: (1) The hydrogels using OSG will show morphological change, increased thermal stability, and improved mechanical properties. (2) As the Schiff base is formed between OSG and acrylamide (AM), the hydrogel will exhibit self-healing properties. (3) The fabricated OSG/P(NIPAM-AM) hydrogels will show temperature sensitivity due to both the temperature sensitivity of the PNIPAM group of P(NIPAM-AM) and the pH responsiveness of the carboxyl groups of OSG. (4) Then, the temperature/pH-responsive drug release effect of OSG/P(NIPAM-AM) hydrogels will be exhibited, as well as biocompatibility. Therefore, the hypothesis was verified by examining the microstructure, swelling behavior, and rheological and mechanical properties of the hydrogel, and the drug release effect according to temperature/pH and cytotoxicity.

## 2. Materials and Methods

### 2.1. Chemicals and Materials

N-isopropylacrylamide (NIPAM) (purity 98%) monomer was purchased from TCI (Tokyo, Japan). Acrylamide (AM) (purity > 99.5%) monomer was purchased from BioShop (Burlington, ON, Canada). Ammonium persulfate (APS) with a purity over 99% and *N*, *N*, *N’*, *N’*-tetramethylenediamine (TEMED) with a purity higher than 98% were purchased from Bio-Rad (Hercules, CA, USA). *N*, *N’*-methylenebis(acrylamide) (MBA) was obtained from Sigma Aldrich (St. Louis, MO, USA). Sodium periodate (NaIO_4_) (purity 98%) was purchased from Alfa Aesar (Ward Hill, MA, USA).

### 2.2. Preparation of Oxidized Succinoglycan (OSG)

Succinoglycan (SG) was obtained directly by culturing, isolating, and purifying the *Sinorhizobium meliloti* Rm1021 strain in glutamic acid-mannitol-salts (GMS) medium containing trace elements at 30 °C for 7 days. The oxidized SG was synthesized using the previously reported sodium periodate oxidation method [21,22]. Briefly, SG (1.0 g, 0.66 mmol) was completely dissolved in 90 mL of distilled water (DW), and 10 mL of sodium periodate solution (720 mg, 3.3 mmol) was added. Then, the mixed solution was stirred and reacted for 5 h at 25 °C in the dark. Then, ethylene glycol was added to the mixture and stirred for an additional 1 h to complete the reaction. Finally, the dialysis membrane was replaced in distilled water for 48 h to remove unreacted chemicals. The extent of oxidation was quantified by measuring the number of aldehydes using the iodometric titration method [23,24]. The characterizations of OSG were described in Appendix A.

### 2.3. Preparation of OSG/Poly (N-Isopropylacrylamide-Co-Acrylamide [OSG/P(NIPAM-AM)] Hydrogels 

To prepare the composite hydrogels, NIPAM, AM, MBA, and various amounts of OSG (Table 1) were dissolved in 10 mL of PBS buffer and stirred overnight. Next, the reaction solution was stirred while bubbling with nitrogen to remove oxygen. Then, APS aqueous solution as an initiator and TEMED as an accelerator were added to perform a copolymerization reaction to prepare hydrogels. The detailed composited hydrogels are shown in Table 1.

### 2.4. Characterization Methods 

#### 2.4.1. Attenuated Total Reflectance Fourier Transform Infrared Spectroscopy (ATR-FTIR)

ATR-FTIR spectra were measured by using a Vertex 80 V FTIR spectrometer (Bruker, Ettlingen, Germany) with a resolution of 0.1 cm^−1^ using 8 scans. The spectra of lyophilized hydrogel samples were recorded in the range from 4000 to 600 cm^−1^.

#### 2.4.2. X-ray Diffraction Analysis (XRD)

The XRD curve of hydrogels was confirmed using a powder XRD diffractometer (SmartLab, Rigaku, Japan) with CuKα radiation in the diffraction angle range of 5–70°.

#### 2.4.3. Thermogravimetric Analysis (TGA)

TGA analysis of the freeze-dried samples was obtained using a PerkinElmer gravimetric analyzer under a flowing nitrogen atmosphere. The prepared samples were measured while heating from 25 °C to 600 °C at a rate of 10 °C/min.

#### 2.4.4. Field Emission Scanning Electron Microscopy (FE-SEM)

After reaching the equilibrium swelling ratio in PBS buffer at room temperature, the swollen hydrogels were quickly frozen in a refrigerator and dried in a freeze-dryer. The lyophilized hydrogel samples were carefully cut, and the cross-sectional morphology was studied by using FE-SEM (Hitachi S-4700, Tokyo, Japan) under an acceleration voltage of 5 kV. The dried samples were coated with a platinum layer at 30 w for 120 s in a vacuum before SEM observation.

### 2.5. Swelling Measurements

Dried hydrogel samples were immersed in PBS buffer and weighed over time in an incubator environment at 25 °C. A conventional weight-swelling ratio was used to determine the degree of swelling of the hydrogels. Temperature and pH-dependent swelling measurements showed that the dried samples swelled in solution at pH (2.0, 7.4) and temperature (25, 37 °C), respectively. Equilibrium swellable hydrogels were weighed after removing surface water with filter paper and calculated according to the following formula [25]:$$Swelling\;ratio\;\left(g/g\right)=\left(W_{s}-W_{d}\right)/W_{d}$$
where *Ws* and *Wd* represent the weight of the swollen hydrogel and dry hydrogel samples, respectively. The measurement was conducted in triplicate.

### 2.6. Rheological Measurements

Rheological measurements were performed in a DHR-2 rheometer (TA Instruments, New Castle, DE, USA) with 20 mm parallel plates measuring at 25 °C. Strain sweep test was conducted within the shear strain range of 0.1–500% at a frequency of 1 Hz. The angular frequency sweep experiment was analyzed from 0.1 rad/s to 100 rad/s at constant 1.0% strain. To further the rheological recovery properties of hydrogels, alternate strain sweep tests were performed at an angular frequency of 1 Hz. The strain amplitude was measured by repeatedly alternating a fixed strain of 0.1% for 100 s and 1000% for 100 s.

### 2.7. Mechanical Test

The compressive and tensile experiments were performed by an Instron E3000LT (Instron Inc, Buckinghamshire, UK). The hydrogels for the tensile tests were manufactured in the form of a cylindrical shape with a diameter of 4 mm and a height of 60 mm, and for the compression test, they were manufactured in the form of a cylinder with a diameter of 14 mm and a thickness of 60 mm. The compressive and tensile tests were performed at the speed of 5 mm/min.

### 2.8. In Vitro Drug Release Measurement

For in vitro drug release studies, 5-fluorouracil (5-FU) was selected as a model drug and loaded into the prepared hydrogels. The drug 5-FU is one of the potential anti-cancer drugs to treat colon, ovarian, uterine, and liver cancers [26]. Because it is a slightly water-soluble pyrimidine antimetabolite, it has a short biological half-life [27]. Therefore, long-term and continuous administration is required to obtain the maximum therapeutic effect. Drug loading was performed at room temperature for 48 h by immersing the dried hydrogels in 5 mL of an aqueous drug solution (1 mg/mL) and swelling. After drying the hydrogel loaded with the next drug, it was stirred in 5 mL of PBS solution for 12 h. The absorbance of the supernatant obtained by centrifugation was then recorded at 265 nm using a UV-Vis spectrophotometer. The drug loading amount and encapsulation efficiency were calculated from the following equations, respectively [28,29]:$$Drug\;loading\;amount\;\left(\frac{\mathrm{mg}}{g}\right)=\frac{Concentration\;of\;5-FU\times 5\;\mathrm{mL}}{Weigh\;of\;hydrogels\;\left(g\right)}$$
$$Encapsulation\;efficiency\;\left(\%\right)=\frac{Actual\;loaded\;drug}{Theoretical\;drug}\times 100\%$$

Based on this formula, the loading of 5-FU in the hydrogel was the highest at 26 mg/g (Appendix A). The drug release experiment was performed with OSG3/P(NIPAM-AM) hydrogel at 25 and 37 °C with constant stirring at 100 rpm. The drug-loaded hydrogels were immersed in drug-releasing solutions (at pH 2.0 and 7.4). At specific interval points, 1 mL of 5-FU release solution was measured at an absorbance of 265 nm. These results were expressed by the following equation [30]:$$Cumulative\;amount\;of\;the\;5-FU\left(\%\right)\;\;\;\;\;\;\;\;\;\;\;\;\;\;\;\;\;\;\;\;\;\;\;\;\;\;\;\;\;\;\;\;\;\;\;\;\;\;\;=\left[C_{n}V+\overset{i=n-1}{\underset{i=1}{\sum }}C_{i}V_{i}/totalencapsulated\;5-FU\;in\;sample\right]\times 100$$
where *C_n_* and *C_i_* represented the concentrations of 5-FU in the release medium and extraction sample, respectively. V and Vi were the release medium volume and sample volume, respectively.

### 2.9. Cytotoxicity Test

Cell viability of the prepared hydrogels was evaluated for toxicity of human embryonic kidney 293 (HEK-293) cells using WST-8 (QuantiMax, Seoul, Republic of Korea) assay. HEK-293 cells were seeded at 2 × 10^4^ per well in a 96-well plate loaded with hydrogel samples. The cells were cultured in Dulbecco’s modified Eagle medium (DMEM, WELGENE, Gyeongsan-si, Republic of Korea) containing 10% fetal bovine (FBS) and 1% antibiotics (100 U/mL penicillin and 100 g/mL streptomycin). The plate was then incubated at 37 °C with 5% CO_2_ air for 48 h. After that, the hydrogels and the culture supernatant were removed, and a new DMEM and WST-8 solution were added and incubated for 4 h. The optical density was determined at a wavelength of 450 nm with a SpectraMax ABS Plus Microplate Reader (Molecular Devices, Sunnyvale, CA, USA). The cell viability was determined using the following equation [31]: $$Cell\;viability\;\left(\%\right)=\left(A_{samples}/A_{Control}\right)\times 100\%$$

## 3. Results and Discussions

### 3.1. Characterization of OSG

Natural SG are oxidized by sodium periodate to transform C2/C3vicinal hydroxyl groups of monosaccharide into dialdehyde groups [20,22]. The oxidation degree of SG was determined by iodine titration, and, as a result, the degree of oxidation was 84.89 ± 0.09% [23,32]. OSG contains succinyl, acetyl, and pyruvyl groups, which are characteristic peaks of SG, and these peaks were analyzed in the ^1^H NMR spectrum of OSG (Appendix A). The characteristic peaks of SG in the ^1^H NMR spectrum indicated the methylene proton of the succinyl group at 2.45 and 2.59 ppm, the methyl proton of the acetyl group at 2.11 ppm, and the pyruvate group at 1.43 ppm. The new peak of aldehyde proton due to SG oxidation was observed at 9.06 ppm [22]. 

ATR-FTIR analysis was also used to confirm the characteristic carbonyl stretching vibration of OSG, and the absorption peaks could be seen in Appendix A. SG showed absorption peaks corresponding to -OH, -CH_2_, and C-O stretching at 3299 cm^−1^, 2889 cm^−1^, and 1040 cm^−1^, respectively. The absorption peak at 1727 cm^−1^ was associated with the C=O stretching of the acetyl ester. The absorption peaks observed at 1613 cm^−1^ and 1373 cm^−1^ are assigned to asymmetric and symmetric COO−stretching vibration of the succinate and pyruvate groups [22]. However, the acetyl group of the carbonyl ester peak was shifted from 1727 cm^−1^ to 1725 cm^−1^ due to the periodate oxidation of SG. In addition, the peak of OSG was observed at 894 cm^−1^ in the ATR-FTIR spectrum, which is attributed to the formation of hemiacetal bonds between aldehyde groups and adjacent hydroxyl groups [21]. 

### 3.2. Formation of OSG/P(NIPAM-AM) Hydrogels

The structure of hydrogels was confirmed by ATR-FTIR characterization, depicted in Figure 1. The spectra of the P(NIPAM-AM) hydrogels showed absorption bands for carbonyl C=O stretching of amide 1, N-H bending of amide 2, and C-N stretching of amide 3 at 1640 cm^−1^, 1541 cm^−1^, and 1171 cm^−1^, respectively. The intensity of the band at 1452 cm^−1^ is attributed to the C-H bending of the -CH(CH_3_)_2_ and –(CH_2_)_2_ groups [33,34]. For the OSG/P(NIPAM-AM) and P(NIPAM-AM) hydrogels, absorption peaks can be seen between 2890 and 3000 cm^−1^, which correspond to the C-H stretching of the -CH_3_ and -CH_2_- groups. The broad absorption band of over 3000 cm^−1^ is assigned to the stretching O-H and N-H bonds [35]. However, as OSG was introduced into the hydrogels network, the peaks of O-H, N-H, amide 1, and amide 2 appear as shifted absorption bands at 3284 cm^−1^, 3189 cm^−1^, 1648 cm^−1^, and 1538 cm^−1^, respectively, in OSG/P(NIPAM-AM). In addition, the glycosidic bond C-O stretching peak of OSG at 1060 cm^−1^ showed a peak shift and reduced intensity. These results demonstrated that OSG was introduced into the P(NIPAM-AM) network, and the hydrogels were successfully synthesized [36,37].

XRD patterns of the composite hydrogels and OSG are shown in Figure 2. The XRD pattern of OSG showed broad diffraction peaks at 2θ = 19°, as well as shifting and broadening, compared to the crystalline peak of SG shown in previous studies [38]. This implied that the structure of SG was changed during oxidation, and the intermolecular hydrogen bonding was weakened as the molecular weight decreased [39]. The XRD pattern of the P(NIPAM-AM) hydrogels has a broad peak at 2θ = 21.8° associated with the polymeric material. Some sharp peaks can be seen in the XRD pattern of the P(NIPAM-AM) at 2θ = 27.4°, 31.9°, 45.7°, and 56.5°, indicating crystalline properties [40]. The XRD diffraction pattern of the OSG/P(NIPAM-AM) composite hydrogels also showed strong signals due to the polymer material, but showed a wider diffraction angle at 2θ = 23.1°. This result indicated that OSG was introduced into the P(NIPAM-AM) network, causing a more significant disorder or detachment [41].

### 3.3. Thermal Gravimetric (TGA) Analysis

TGA was performed to study the decomposition pattern and, hence, the compositional differences of the samples. The thermogravimetric (TG) and derivative thermogravimetric (DTG) curves of OSG and hydrogel samples are shown in Figure 3. In the TGA graph in Figure 3a, all samples showed weight loss near 100 °C, which corresponds to moisture loss. The tendency for mass loss to increase as the concentration of OSG increases is attributed to the carboxyl group of SG [42]. OSG showed two rapid mass losses between 200 and 500 °C, and 93% mass loss resulted at 600 °C. This curve of OSG occurred because the molecular weight decreased as the SG was oxidized [43]. For the P(NIPAM-AM) sample, the primary decomposition stage was observed at 300–400 °C, and the secondary decomposition was 600 °C, with a maximum mass loss of 77%. This result is related to the degradation of the backbone of the polymer-polymer interaction hydrogels [44]. Between 200 and 400 °C, the TGA curves for OSG1/P(NIPAM-AM) and OSG2/P(NIPAM-AM) showed a similar mass loss pattern to the P(NIPAM-AM) gel. In the case of OSG3/P(NIPAM-AM) and OSG4/P(NIPAM-AM), the higher the ratio of OSG in the 200–400 °C range, the more mass loss was shown. This indicated strong intermolecular interaction between copolymer and OSG. However, at 400 °C or higher, the OSG4/P(NIPAM-AM) gel degrades slowly, while the remaining hydrogel maintains the degradation rate. Thus, the composite hydrogel exhibited a higher residual mass with increasing OSG content, indicating good thermal stability [20,21]. 

Figure 3b shows the DTG curves of the prepared hydrogels with a scanning range from 25 to 600 °C. The maximum weight loss peak at 389 °C in the P(NIPAM-AM) gel is due to the degradation of the polymer-polymer interaction chain of copolymer hydrogels [44]. In the DTG curve, the decomposition temperature of composite hydrogels, such as OSG1/P(NIPAM-AM) (381 °C), OSG2/P(NIPAM-AM) (383 °C), OSG3/P(NIPAM-AM) (377 °C), and OSG4/P(NIPAM-AM) (390 °C), changed depending on the concentration of OSG. These results could be explained by the interaction between the OSG and the copolymer network [45,46]. As the drug molecules are loaded into the hydrogel, it shows higher thermal stability, suggesting that it can be applied as a drug delivery system because of its thermally induced release (Appendix A).

### 3.4. FE-SEM Morphology

The result of observing the cross-sectional structure of the hydrogels with a scanning electron microscope is displayed in Figure 4. The tunable pore size of hydrogels affects drug loading and cell proliferation for drug delivery and tissue engineering [47,48]. The dried hydrogel sample exhibited a uniformly cross-linked 3D porous structure. The P(NIPAM-AM) gel showed small-sized closed pores by crosslinking MBA and high-density polymers. OSG1/P (NIPAM-AM) was more severe, and these results also affect cell viability (Section 3.8). For composite hydrogels, as the OSG content of the OSG/P(NIPAM-AM) hydrogels increased, a more uniform 3D structure was shown, and the average pore size also gradually increased, as shown in Figure 4b–d. This may be related to the formation of large pores by absorbing more moisture by repulsive force, as the number of carboxylate groups in the hydrogel increases as the content of hydrophilic OSG increases [49]. The 5-FU-loaded hydrogels still display porosity and networks, indicating drug delivery by swelling (Appendix A) [50,51].

### 3.5. Rheological Properties of Hydrogels

The rheological properties of hydrogels were evaluated through the strain and frequency sweep tests. Storage modulus (G’) and loss modulus (G”) determine the viscoelasticity of a hydrogel as the amount of energy stored and dissipated under vibrational stress, respectively [52]. Oscillation angular frequency sweep tests investigated the viscoelastic properties of hydrogels with different OSG contents. As shown in Figure 5a, the G’ of all samples was higher than the G”, which means that the hydrogel was in a solid gel state [53]. The G’ value for P(NIPAM-AM) was 3373 Pa; for the OSG4/P(NIPAM-AM) gel, it was 831 Pa, which was about a 4-fold decrease compared to P(NIPAM-AM). This is because the gel flexibility was improved by introducing OSG into the P(NIPAM-AM) network [25]. The loss tangent (G”/G’, tan δ) curve shows the relative contribution of the viscous component of the hydrogel as a strain function. Figure 5b is a loss tangent curve showing a function of the strain of the hydrogel. The tan δ values of all samples increased with increasing OSG concentration. Thus, it explained that the flexibility difference occurred as the chains of OSG increased in the hydrogel network [54]. 

The strain sweep tests were performed at a frequency of 1.0 Hz from 0.1% to a strain of 1000% to determine the limits of the linear viscoelastic region for hydrogel samples. As shown in Figure 5c, all hydrogel samples showed linear behavior in the strain range from 0.1 to 250%, but when it was over 250%, G’ decreased sharply, indicating the intersection of G’ and G”. The curves of each hydrogel showed a crossover point at strain 298% (P(NIPAM-AM)), 400% (OSG1/P(NIPAM-AM)), 395% (OSG2/P(NIPAM-AM)), 432% (OSG3/P(NIPAM-AM)), and 813% (OSG4/P(NIPAM-AM)) (Figure 5d). This means that each sample experienced the collapse of the hydrogel network in the corresponding strain and that the critical deformation intersection of the hydrogel gradually increased as the concentration of OSG increased [55]. Although P(NIPAM-AM) is a relatively hard and brittle hydrogel, it can cause more flexible hydrogels with higher concentrations of OSG. 

The damage healing of hydrogels with post-damage recovery properties of networks at large amplitude oscillatory strains and low strain was assessed (Figure 6a). A 1000% size strain was applied to damage the OSG/PNIAPM hydrogels for 100 s. After that, a strain of 1% was applied for 100 s to confirm the recovery of the hydrogel network. The strength value was restored immediately after changing from large strain to low strain. In addition, the hydrogels recovered almost instantaneously whenever the relieved strain was applied, and the recovery process was observed to be nearly consistent for three alternating cycles. Self-healing properties can be qualitatively demonstrated by cutting different colored hydrogels into two pieces and then making quick contact between two different colored parts. For better visualization of self-healing, two samples printed in different colors, red and green, were cut and rejoined (Figure 6b). As shown, the structure has been macroscopically repaired

### 3.6. Mechanical Properties of Hydrogels

To further elucidate the mechanical behaviors of the OSG/P(NIPAM-AM) hydrogels, tensile and compressive stress-strain tests were performed. As shown in Figure 7a, the P(NIPAM-AM) hydrogels exhibited a maximum compressive stress of 582 kPa at a breaking strain of 95%. The compressive strength of OSG/NIPAM hydrogels at 95% compressive strain was OSG1/PNIPAM (582 kPa), OSG2/PNIPAM (339 kPa), and OSG3/PNIPAM (192 kPa), and OSG4/PNIPAM (215 kPa). For OSG2/P (NIPAM-AM), the gel network was fractured at 75% strain. Eventually, the compressive strength of OSG/P(NIPAM-AM) hydrogels decreased with increasing OSG concentration. The change in the mechanical properties of the hydrogel is due to the co-occurrence of chemical and physical crosslinking and the interpenetration of OSG chains and P(NIPAM-AM) networks [56]. As the OSG concentration increases, the gel network becomes looser as OSG chains into P(NIPAM-AM); thus, the compressive strength of the gel is relatively low.

The tensile stress-strain curve can be seen in Figure 7b to confirm the tensile fracture stress of the OSG/P(NIPAM-AM) hydrogels. The fracture strain of the P(NIPAM-AM) hydrogel was 88%, with a tensile strength of 66 kPa. The OSG1/P(NIPAM-AM) hydrogel had a tensile strength of 84 kPa and a strain of 114%, which slightly improved the mechanical strength compared to the P(NIPAM-AM) gel. OSG2/P(NIPAM-AM) had the highest strain fracture point at 600%, but the tensile strength was 260 kPa. OSG3/P (NIPAM-AM) exhibited the highest tensile strength (260 kPa) with breaking strain (146%). However, in the case of the OSG4/P(NIPAM-AM) hydrogel in the tensile curves, the higher the OSG concentration, the weaker the tensile strength. Although the introduction of OSG loosened the hydrogel network, the hydrogen bonding between OSG and the P(NIPAM-AM) chains allows it to withstand higher tensile stress [57].

### 3.7. Swelling Behavior of Hydrogels

To confirm the swelling behavior of OSG/P(NIPAM-AM) hydrogels, we performed swelling kinetics experiments with time, temperature, and pH. From the results of the swelling kinetics over time, all hydrogels swelled rapidly during the first 24 h and then slowly reached equilibrium swelling at a low rate (Figure 8a). OSG/P(NIPAM-AM) hydrogels showed an increase in the equilibrium swelling ratio with increasing OSG content. Among them, the OSG3/P(NIPAM-AM) hydrogels showed a 1.2-fold higher swelling rate than the P(NIPAM-AM) gel. This phenomenon means that as the content of hydrophilic OSG increases, many hydrogen bonds are formed between the hydrogels and water molecules, thereby increasing the swelling rate [21].

In general, the LCST of PNIPAM is about 32 °C in an aqueous solution, and it is well known that the LCST increases or decreases depending on the amount or combination of hydrophilic and hydrophobic monomers [42]. Compared with the DSC results of PNIPAM published in a previous study, the LCST of P(NIPAM-AM) hydrogel shifted to a higher temperature, and the temperature sensitivity of the hydrogels increased with the introduction of OSG (Appendix A). As can be seen in Figure 8b, all hydrogels have a higher swelling ratio at 25 °C than at 37 °C in pH 7.4 buffer solution. However, OSG3/P(NIPAM-AM) has a relatively large difference in swelling ratio at the two temperatures. Similar results have also been reported in other studies [58]. 

Figure 8c displayed the change in the swelling behavior of the hydrogel at various temperatures and pH. In the case of PNIPAM, there was no significant difference in the swelling ratio between the two pH solutions, and there was only a slight difference between the two temperatures. Unlike the PNIPAM gel, the OSG-introduced hydrogel showed a higher degree of swelling at high pH values. Because the pKa of the succinyl group in OSG was 3.8, most of the carboxylic acid groups at high pH were in the COO^-^ form. In a pH 7.4 buffer, the ionized carboxylic acid groups and water molecules interact by hydrogen bonding to increase the swelling ratio [49,59]. Therefore, the change in swelling of the OSG3/P(NIPAM-AM) hydrogels suggests a temperature- and pH-responsive drug delivery effect. The results obtained from the FE-SEM analysis further supported the results for the temperature and pH dependence of OSG3/PNIPAM hydrogels (Appendix A).

### 3.8. Drug Loading and Release

Drug loading of 5-FU was manufactured by the hydrogel swelling method. As shown in Appendix A, in the results of measuring the drug-loaded hydrogels, the OSG3/P (NIPAM-AM) drug load was the highest at 26 mg/g. The interaction between drug and hydrogels was probably due to the hydrogen bonding between the functional groups (COOH and OH) of succinoglycan and 5-FU functional groups (NH and CO) inside the hydrogel. Thus, an in vitro drug release study was conducted using OSG3/P (NIPAM-AM) hydrogels. The results of examining the effects of temperature and pH on the cumulative release of 5-FU from OSG3/P (NIPAM-AM) hydrogels are shown in Figure 9. Figure 9a shows the temperature-response cumulative release curves at 25 °C and 37 °C of the release hydrogels sample in pH 7.4 buffer. The 5-FU release of OSG3/P(NIPAM-AM) hydrogel in pH 7.4 buffer was significantly higher at 37 °C than at 25 °C. It was approximately 65% at 37 °C and 52% at 25 °C, with the sustained release for 120 h. At 37 °C, the hydrogel contracts as the hydrophobic interactions of the copolymer chains are enhanced, thereby releasing the drug [36]. Figure 9b shows the difference in drug release in pH 7.4 and pH 2.0 buffers when the release temperature is 37 °C. In this case, the hydrogel released 54% at pH 2.0 and 65% at pH 7.4, and this difference can be attributed to the swelling inside the hydrogel induced as a result of the deprotonation of the succinyl group of OSG at pH 7.4. Therefore, it has been shown that temperature/pH-responsive SG-based hydrogels have effectively controlled drug release behavior in the simulated stomach (pH 2.0) and intestinal fluid (pH 7.4). These properties show the promising applications of OSG3/P(NIPAM-AM) hydrogels in stimuli-responsive drug delivery.

### 3.9. Cell Viability of Hydrogels

Low cytotoxicity is the priority for medical materials in drug delivery in organisms. From the results observed in Figure 10, the cell viability of P(NIPAM-AM) hydrogel was 105%. The cell viability of HEK-293 cells was 81% (OSG1/P(NIPAM-AM), 111% (OSG2/P(NIPAM-AM), 112% (OSG3/P(NIPAM-AM), and 121% (OSG4/P(NIPAM-AM) for OSG/PNIPAM hydrogels. The cell viability of DMSO-treated cells as a positive control was 15%. The cell viability of OSG1/P(NIPAM-AM) gel was 81%, which showed low cell viability, but within the acceptable range of 75–85% [37]. The FE-SEM results suggested that the dense pores of the OSG1/P(NIPAM-AM) hydrogel prevented cells from moving to the structural center, which affected nutrient diffusion and waste removal [60]. The hydrogel viability results for HEK-293 cells confirmed that the OSG/P(NIPAM-AM) hydrogel is non-toxic, has good biocompatibility, and can be a scaffold-like material capable of drug delivery. In addition, the hydrogels also showed biodegradability (Appendix A). 

## 4. Conclusions

We successfully fabricated multifunctional OSG/P(NIPAM-AM) hydrogels with mechanical strength, swelling behavior, and temperature/pH-responsive drug release effects using OSG and copolymers. OSG was obtained through periodate oxidation of SG, and its structural analysis was performed. After confirming the structure, OSG was introduced into the P(NIPAM-AM) network to synthesize OSG/P(NIPAM-AM) hydrogels, and chemical structure and physical properties were confirmed through ATR-FTIR, XRD, TGA, and FE-SEM. The OSG/P(NIPAM-AM) hydrogels with an increased pore size showed a relatively high swelling ratio as the OSG concentration increased, compared to the P(NIPAM-AM) gel, and showed a difference in the swelling degree as the temperature increased. The resulting OSG/P(NIPAM-AM) had a 4-fold greater improvement in compressive strength than P(NIPAM-AM) and showed flexibility and self-healing ability. In addition, the 5-FU-loaded OSG/P(NIPAM-AM) hydrogel showed release characteristics in response to temperature and pH. Cytotoxicity tests revealed that the OSG/P(NIPAM-AM) hydrogels were non-toxic. Overall, self-healing, temperature/pH-responsive multifunctional OSG/P (NIPAM-AM) hydrogels could potentially be used in biomedical applications, such as drug release, tissue engineering, intelligent scaffolds, and wound healing.

## Data Availability

Not applicable.

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
