# Peer review of "Multifunctional Oxidized Succinoglycan/Poly(N-isopropylacrylamide-co-acrylamide) Hydrogels for Drug Delivery"

_polymers, 2022, doi:10.3390/polym15010122_

Round 1

Reviewer 1 Report

Authors fabricated drug loaded oxidized succinoglycan/P(NIAM-co-AM) hydrogels for temperature/pH responsive drug delivery systems. The system was characterized by FTIR, TGA, DSC and FE-SEM, tensile strength and rheological measurements. Cytotoxicity of the drug loaded hydrogel was also studied. This manuscript attracts readers in the field of polymer based drug delivery system. However, major revision of the manuscript needed before it is accepted.

1. The drug loading efficiency of OSG3 was 26 mg/g. What is weight of hydrogel (dry) used for drug loading. Calculate the encapsulation efficiency percentage.

2. Why there is a difference in slope value at pH 2.0 and 7.4? 

3. Authors should include TGA, DSC and FESEM for drug loaded OSG/ P(NIAM-co-AM) hydrogel samples too and discuss the results.

4. The interaction between drug and hydrogel needs to be explained.

5. In page 9, line 295, “OSG10/P (NIPAM-AM) was more severe” please check the sample name OSG10.

6. Section 3 title should be Results and Discussion.

Reviewer 2 Report

Dear Authors

The presented work is very interesting for the readers.

Self-healing and temperature/pH-responsive hydrogels using oxidized succinoglycan (OSG) and a poly (N-isopropyl acrylamide-co-acrylamide) [P(NIPAM-AM)] copolymer were developed. OSG was synthesized by periodate oxidation of succinoglycan (SG) isolated directly from soil microorganisms, Sinorhizobium meliloti Rm1021. The OSG/P(NIPAM-AM) hydrogels were obtained by introducing OSG into P(NIPAM-AM) networks. The chemical structure and physical properties of these hydrogels were characterized by ATR-FTIR, XRD, TGA, and FE-SEM. The OSG/P(NIPAM-AM) hydrogels showed improved elasticity, increased thermal stability, new self-healing ability, and 4-fold enhanced tensile strength compared with the P(NIPAM-AM) hydrogels. Furthermore, the 5-FU-loaded OSG/P(NIPAM-AM) hydrogels exhibited effective temperature/pH-responsive drug release. Cytotoxicity experiments showed that the OSG/P(NIPAM-AM) hydrogels were non-toxic, suggesting that OSG/P(NIPAM-AM) hydrogels could have the potential for biomedical applications such as stimuli-responsive drug delivery systems, wound healing, smart scaffolds, and tissue engineering.

Indeed, Some points need to declare and others need to be studied.

For example, biodegradability and biocompatibility have to be studied in addition to cytotoxicity.

The cytotoxicity study used only the lowest OSG/P(NIPAM-AM) hydrogel composite (OSG1), while the best results of the temperature and pH sensitivity in addition to the drug-controlled release was obtained using OSG3/P(NIPAM-AM) hydrogel.

The SEM study of the OSG3/P(NIPAM-AM) hydrogel after drug release needs to monitor. 

The authors have to declare the reason for choosing 5-FU as a drug model and mentioned it in the materials section with specifications.

 In conclusion, a major revision is needed before considering the manuscript for publication. 

Round 2

Reviewer 1 Report

The manuscript was revised as per the reviewer comments and now can be accepted in present form for publication

Reviewer 2 Report

Dear Authors

The revised version has taken into consideration the raised comments and added further experimental results based on the request. Furthermore, the additional added results correlate to the achieved goal of the manuscript.

I can recommend the current version of the manuscript for publication.